# Efficacy of Manual Therapy and Transcutaneous Electrical Nerve Stimulation in Cervical Mobility and Endurance in Subacute and Chronic Neck Pain: A Randomized Clinical Trial

**DOI:** 10.3390/jcm10153245

**Published:** 2021-07-23

**Authors:** Belén Díaz-Pulido, Yolanda Pérez-Martín, Daniel Pecos-Martín, Isabel Rodríguez-Costa, Milagros Pérez-Muñoz, Victoria Calvo-Fuente, María Félix Ortiz-Jiménez, Ángel Asúnsolo-del Barco

**Affiliations:** 1Physiotherapy Department, University of Alcalá, 28871 Madrid, Spain; belen.diazp@uah.es (B.D.-P.); daniel.pecos@uah.es (D.P.-M.); isabel.rodriguezc@uah.es (I.R.-C.); milagros.perez@uah.es (M.P.-M.); victoria.calvo@uah.es (V.C.-F.); 2Puerta de Madrid Health Center, Public Health System of the Community of Madrid, 28802 Madrid, Spain; mfortiz2@yahoo.es; 3Surgery, Medical and Social Sciences Department, University of Alcalá, 28871 Madrid, Spain; angel.asunsolo@uah.es

**Keywords:** musculoskeletal manipulation, neck pain, physical therapy modalities, public health, transcutaneous electric nerve stimulation (TENS)

## Abstract

Neck pain is a frequent health problem. Manual therapy (MT) and transcutaneous electrical nerve stimulation (TENS) are recommended techniques for treatment of mechanical neck disorders (MND) in Spanish Public Primary Care Physiotherapy Services. The aim of this study was to compare the efficacy of MT versus TENS in active mobility and endurance in cervical subacute or chronic neck pain. Ninety patients with MND were randomly allocated to receive ten 30-min sessions of either MT or TENS, in a multi-centered study through 12 Primary Care Physiotherapy Units in the Madrid community. Active cervical range of motion (CD-ROM) and endurance (Palmer and Epler test) were evaluated pre- and post-intervention and at 6-month follow-up. A generalized linear model of repeated measures was constructed for the analysis of differences. Post-intervention MT yielded a significant improvement in active mobility and endurance in patients with subacute or chronic MND, and at 6-month follow-up the differences were only significant in endurance and in sagittal plane active mobility. In the TENS group, no significant improvement was detected. With regard to other variables, MT improved mobility and endurance more effectively than TENS at post-intervention and at 6-month follow-up in the sagittal plane. Only MT generated significant improvements in cervical mobility and endurance in the three movement planes.

## 1. Introduction

Nowadays neck pain is a frequent health problem associated with modern lifestyles [1], and is thought to affect between 21 and 71% of the population at some point in their lives [2]. Between 30% and 50% of the adult population reports having experienced neck pain in the previous year [3], and 50% to 85% of patients with neck pain will report neck pain 1 to 5 years later [4]. Musculoskeletal disorders, including chronic neck pain, produce increasing problems for patients and national economies [2,3,5,6,7,8,9,10,11,12,13,14]. The themes of disability deriving from chronic pain of the spine and appropriate approach to its treatment are current research topics in this field [15].

The annual prevalence of neck pain with disability, but without structural damage, varies between 1.7% and 11.5% in the general population [16]. It has been estimated that in Spain, chronic neck pain affects 9.6% of men and 21.9% of women [17] (D), and about 10% of the cases referred to physical therapists are due to neck pain [18].

Although treatments such as nonsteroidal anti-inflammatory drugs and dry needling are usually first-in-line in the field of referred pain, such as orofacial pain [19], the effects of conservative interventions in the treatment of neck pain, which include manual therapy (MT) and electrotherapy among others, have mainly been studied post-intervention with inconclusive results [6,20,21,22,23,24,25,26]. MT and transcutaneous electrical nerve stimulation (TENS) are recommended techniques for the treatment of mechanical neck disorders (MND) in Spanish Public Sector Primary Care Physical Therapy Services [27,28,29,30]. In this framework, both therapies are accepted as standard clinical practice, and the choice of one or the other depends on the physical therapist’s decision based on clinical patient manifestations, personal experience and formation, and time and available resources.

MT intervention, which includes manipulation, mobilization, massage, and neuromuscular techniques [31], has not been proven to be effective when used in an isolated way to improve mobility and muscular endurance in MND [6,8,20,21,26,31,32,33,34]. Whereas the Ottawa Panel was able to demonstrate that massage interventions are effective for relieving immediate post-treatment neck symptoms of pain, tenderness, and range of motion, data on the long-term effects are insufficient [2]. It has been proven that MT combined with pharmacological treatment, exercises, and medical advice is effective post-intervention [34]. Several studies [4,6,9,13,14,34,35,36,37] indicate that MT combined with exercise is more effective than other therapies or placebo to improve symptoms or to increase the patient’s satisfaction, although more estimates are necessary to understand the magnitude of the effect [4,9,20,24,33,34,37,38,39].

Regarding the use of electrotherapy, there have been studies of clinical trials with a small number of patients with MND that compared TENS with other treatments, without finding differences between groups post-intervention [6,8,9,31,33,40,41]. The results of studies on the effectiveness of TENS are scarce, limited, or contradictory [33,37]. Even the Royal Dutch Society for Physical Therapy indicates that it is not recommended that clinicians use electrotherapy for patients with neck pain grade I or II [4].

On the other hand, biomechanics and psychological factors have been identified in the transition from an acute to a chronic pain syndrome. In MND without an injury, it is considered that chronification is due to biomechanical overloading. If pain relief is the only aim of the treatment, then functional disorders (lack of strength, deficit of mobility, posture disorder, etc.) can persist and be precursors of future deficiency biomechanics and chronification of MND [42,43,44].

No study that compares the efficacy of both techniques on mobility and endurance has been found. The aim of the present study was to compare the effects of MT and TENS techniques on cervical active mobility and muscle endurance in patients with subacute and chronic mechanical neck disorders. A research of pain intensity relief has been previously published [45]. It was hypothesized that patients who received MT would have greater improvement in cervical mobility and endurance than those who received TENS.

## 2. Materials and Methods

### 2.1. Study Design

A multi-centered randomized clinical trial was performed in order to compare the effectiveness of MT versus TENS in patients diagnosed with subacute and chronic MND without neurologic signs according to the Quebec Task Strength Classification on Spinal Disorders [46] (grades I and II, according to 2000–2010 Neck Pain Task Force (NPTF) classification [16]) and neck pain with mobility deficits, neck pain with movement coordination impairments, and neck pain with headache (cervicogenic), according to the Neck Pain Revision [37]. The present study showed the effectiveness of MT and TENS in improving mobility and endurance.

The clinical trial was approved by the Ethics Committee of Hospital Clínico Universitario San Carlos. It was registered as NCT01153737 at www.ClinicalTrials.gov and funded by Instituto de Salud Carlos III, Fondo de Investigación Sanitaria/Fondos Europeos de Desarrollo Regional (PI/041320), (Spain).

### 2.2. Patients

Subjects included were patients between 18 and 60 years old, who were treated in 12 primary health care physiotherapy units that took part in the study in the Madrid region. Exclusion criteria were as follows: signs of neurological damage, pregnant women, previous neck rachis surgery, patients who received physical therapy or an alternative treatment of the neck or shoulder 6 months prior to the beginning of the study, or those with important psychiatric disorders or other health problems that would contraindicate the techniques to be used (i.e., pacemaker). Patients with neck pain caused by an inflammatory, neurological, or rheumatic disease, severe osteoporosis, fracture, luxation, or vertebrobasilar insufficiency were also excluded.

Consecutive patients with chronic neck pain referred by their primary care physicians and who met the inclusion criteria (revised by the primary care physical therapists) and signed the informed consent were recruited. Blinded allocation, concealed in closed envelopes, was on the basis of block randomization and was carried out also by the primary care physical therapists. Random sequences of 6 patients, assigned by external researchers, were obtained using the statistical program Epidat 3.1© (Xunta de Galicia, Santiago de Compostela, Spain), in order to create 2 equivalent groups.

Patients were evaluated pre-intervention, post-intervention, and at 6-month follow-up. The evaluations were assessed by a different group of physical therapists, so outcome variable evaluation was blinded. A detailed description of the trial’s protocol has been published previously by the investigation group [47].

### 2.3. Sample Size Determination

Ninety patients from the multicenter randomized clinical trial were included. The final potency of the study to detect clinically relevant changes (10/100 points) with the neck disability index (NDI) was 91.4%. Sample size determination was calculated according to the NDI, because data related to clinical significance of active mobility and cervical endurance were not found.

Based on the consulted bibliography, if the neck pain has a muscle-skeletal origin, a clinically relevant result appears when there is a change over 5/50 points in the NDI [48]. In this study, patients with neurological damage were excluded, and the clinical significant change considered was 10/100 points. Taking into account 12 SD points expected in the groups [26,49,50], and with an alfa signification of 0.05 and Beta error of 0.2, 46 patients were necessary, 23 in each group, to detect a clinically relevant difference between them. Ninety patients were required in order to investigate effectiveness in reducing pain intensity. Therefore, all patients from the multicenter randomized clinical trial were included.

### 2.4. Interventions

Therefore, TENS and MT interventions each consisted of 10 sessions, provided by primary care physical therapists for 30 min on alternate days. Applied MT techniques included those which in the Primary Care Physical Therapy Clinical Guidelines. These techniques were regularly used by most of the physical therapists in their daily clinical practice. Figure 1 shows the different components of both interventions.

In this way, the MT techniques consisted of the following: neuromuscular technique, post-isometric stretching, spray and stretching technique, and Jones technique [20,49,50]. These MT techniques are described in Table 1. An expert group, composed of 5 professionals with over 5 years’ experience each in MT physical therapy, decided to include the muscular groups that are regularly treated in this impaired group based on clinical experience, so the following muscles were assessed and selectively treated: superior trapezius, inferior trapezius, levatoranguliscapulaes, sternocleidomastoideus, suboccipital, anterior scalenus, multifidi, subclavie, pectoralis minor, and triceps braquii. In TENS group, a portable digital TENS model TENSMED911 (Enraf-Nonius, Vareseweg, Rotterdam, The Netherlands) was used, with a frequency of 80 Hz, with ≤150 µs pulse duration and adjusted amplitude. TENS electrode placements were located in the painful area, in the metamere, or in the nerve’s pathway [18]. All patients were taught how to perform isometric and neck mobility exercises in their homes and develop postural care skills. This information was explained individually in the first 2 sessions, and each patient also received written information.

During the planning of the study, the physical therapists received 1 session of training to assure homogeneity between the different interventions. The physical therapists that performed the interventions did not have knowledge of the patients’ evaluation data. Every unit received the necessary material (portable digital TENS, cervical range of motion (CROM) instrument, basic inclinometer).

### 2.5. Outcome Measures

Outcome measures of the study were the active range of motion (AROM) and the endurance of the neck muscles, which were registered at first in the 6 movements of the cervical spine, and after this were recorded in the 3 space planes, adding up the degrees of movements in each plane.

Neck mobility was measured with an inclinometer, the CROM Basic Cervical Range of Motion Instrument (Performance Attainment Associates, Sutton-in-Ashfield, UK) [53], which is recommended for active cervical mobility measures in patients with nonspecific neck pain, based on its best scores for clinimetric properties compared to other measurement systems, and also for its ease of use [38].

Endurance of the cervical muscles was measured using the Palmer and Epler test [54]; it is an ordinal categorical variable graduated as follows: “1” not functional, “2” poorly functional, “3” reasonably functional and “4” functional, using the average of the scores obtained in the 2 movements in each plane. It has been proven that, in patients with neck pain, cervical muscle endurance quantifications have excellent intra-observer reliability and moderate inter-observer reliability [55,56].

Other studied variables (descriptive, prognostic, or control) that could have an influence on active mobility and endurance were age, gender, physical disability (according to the Spanish translation of NDI [57], in which a 0 score indicates no disability and 50 the highest disability), pain intensity (measured in millimeters (mm) using the Visual Analogue Scale (VAS) [25]), duration of present neck pain (days), previous neck pain episodes (yes/no), previous accident incurring injury to the cervical spine (yes/no), and fulfillment of the recommended exercises and postural skills (yes/no).

Evaluations were carried out by physiotherapists from Department of University of Alcalá, who were blinded to the groups to which the patients were assigned. A training session was given to the physical therapists´ evaluation group.

### 2.6. Subject Follow up

Three evaluations (pre-, post-intervention, and at 6-months’ follow-up) were performed by physical therapists who disclaimed treatment of each patient. To minimize losses, patients who did not assist in the revision session were phoned at least twice.

Quality control was performed by the coordinating center (Primary Care Research Unit), with periodic supervision and feedback on the study process and data entry; progress reports were performed every 2 to 4 months and meetings with the research physical therapist, every 6 months.

### 2.7. Data Analysis

Statistical analysis was performed without revealing which treatment each patient had received. First, a descriptive analysis of the different studied variables in each intervention group was performed, in order to verify the homogeneity between both groups by using statistical tests.

Analysis of the intervention’s effectiveness was evaluated by protocol comparing the mean value of the answer variables in each of the space planes between both intervention groups pre-intervention, post-intervention, and at 6 months’ follow-up intervention. For this purpose, a general linear model of repeated measures was used, performing a simple contrast, where the reference value was the measurement at the beginning of the study (pre- intervention).

Finally, several models of multiple linear regression were carried out for the mobility and endurance: 1 model for each of the space planes, 1 and 6 months after the intervention.

Variables that were significant in the bivariate analysis and the variables that could cause a confounding phenomenon or interaction were added to the models. It was considered that adjustment was necessary to correct confounding bias if the change in the regression coefficient between the adjusted and unadjusted effect was over 10%. The interaction was studied with statistical significance tests for the parameters, and the interaction terms that were statistically significant (*p* < 0.05) were kept in the model.

Once the maximum model was defined, a gradual inclusion process by steps was performed and guided by the investigators, establishing as a criterion the square value of the adjusted multiple coefficient of determination (R^2^) and Mallow´s Cp. At all times the hierarchy principle was sustained. The statistical program SPSS© 17.0 (Chicago, IL, United States) was used to process and analyze the data.

## 3. Results

In this study 47 and 43 patients were included in the MT and TENS group, respectively, between May 2005 and May 2007. Seventy-two of the enrolled patients finished all interventions (Figure 2).

The mean duration of the current episode of neck pain was 147 days, with a 95% confidence interval (CI) of 94–200 days. Overall, 71 patients (80%) completed the follow- up measurement at 6 months, and the number of losses was similar in both groups.

Sociodemographic characteristics and prognostic variables at baseline in each of the intervention groups, (no significant differences were found in these variables at the beginning of the treatment between both intervention groups, except for endurance in frontal plane (*p* = 0.03 U Mann-Whitney Test)), are shown in Table 2.

In the MT group, statistically significant differences were observed when studying the active cervical mobility and the endurance post-intervention in the three space planes. In this same intervention group, at 6-month follow-up significant differences were observed in the endurance; however, in active cervical mobility, differences were observed only in the sagittal plane. In the TENS group, no significant differences were found post-intervention or at 6-month follow-up. When comparing the mobility and endurance improvement between both intervention groups, no significant differences were found (Table 3). Evolution time is included in Figure 3 and Figure 4. Figure 3 shows little difference between groups in transversal and frontal plane active mobility. Improvement in sagittal plane mobility was maintained 6 months post-intervention in the TM group, but not in the TENS group. Figure 4 shows that the endurance improvement in the TM group was larger than in the TENS group post-treatment and 6 months later.

However, when MT and TENS effects were measured in the presence of other variables, it was found that MT provided more improvement than TENS in mobility and endurance at post-intervention and 6-month follow-up in the sagittal plane. The type of disability was the variable that influenced the results more, regardless of the type of intervention, so the higher the scores were on the NDI at baseline, the better the improvement, especially for mobility at 6-month follow-up. Post-intervention mobility improvements were observed in patients who developed the postural skills, while those patients with longer duration episodes and previous episodes obtained poorer results (Table 4).

Low numbers of adverse effects were observed among patients in the study (6.7% in the MT group and 2% in the TENS group post-intervention, and 2.7% in the MT group and 5.9% in TENS group at 6-month follow-up). Even though at 6-month follow-up adverse effects in the TENS group were double those in the MT group, statistically significant differences were not found at both time points in the study (post-intervention and at 6-month follow-up).

## 4. Discussion

Results of this study showed that post-intervention MT significantly improved active mobility and endurance in patients with subacute or chronic MND, but at 6-month follow-up the differences were only significant in endurance and in sagittal plane active mobility. In the TENS group, no significant improvement was detected either post-intervention or at 6-month follow-up. When measuring the effect of the interventions (MT or TENS) considering other variables, MT improved the mobility and endurance post-intervention and at 6-month follow-up in the sagittal plane to a greater extent than TENS.

It is important to point out that between the confounding and/or modifying variables of our study, it seemed that pain at baseline did not have as important an influence on the outcome measures as the physical disability measured with NDI, independently of the type of intervention, especially at 6-month follow-up. The higher score on the NDI, the larger the increase of mobility after the intervention, which meant that from a worst functional baseline, there seemed to be greater potential for recuperation and a larger range of movement obtained.

The latest and rigorous Neck Pain Clinical Practice Guidelines [4,14,37] propose, based on scientific evidence, offering multimodal care (based primarily on mobilization and/or manipulation in combination with exercise therapy) or stress self-management in patients with persistent or chronic neck pain: thoracic manipulation and cervical manipulation or mobilization; manipulation with soft tissue therapy; high-dose massage; supervised group exercise; supervised yoga; supervised strengthening exercises or home exercises; mixed exercise for cervical/scapulothoracic regions (neuromuscular exercise, stretching, strengthening, endurance training, aerobic conditioning, and cognitive affective elements).

In the same way, for subacute or normal recovery neck pain, they suggest offering multimodal care, manipulation or mobilization, range-of-motion home exercise, or multimodal manual therapy [14], as well as neck and shoulder girdle endurance exercises and thoracic and cervical manipulation and/or mobilization [37]. In case of a normal recovery, the Netherland Guidelines suggest that management should be hands-off, and patients should receive advice from the physical therapist and possibly some simple exercises to supplement “acting as usual” [4].

In no case did these Guidelines recommend the use of TENS based on scientific evidence with indications comparable to the findings of the present study. On the other hand, a recent meta-analysis of TENS for relief of spinal pain published by Resende et al. [58] concluded that the analysis of adequate stimulation parameters for this technique were not significantly different, and that there was no effect on disability; this review provides inconclusive evidence for TENS benefits because of the low quality of the studies and lack of uniformity in parameters and timing of assessment. In the same way, the recent umbrella Review of Systematic Reviews on the efficacy of TENS in cervical pain syndromes published by Paolucci et al. [59], indicates that it was not possible to provide precise recommendations in this regard, and that it is desirable to carry out further studies that support the effectiveness of using TENS in patients suffering from acute and chronic neck pain.

In this same sense, Häkkinen et al. [35] compared MT with neck stretching exercises at home in patients with chronic neck pain, finding a significant increase in active neck mobility in the three space planes 1 month after starting the intervention in both groups, with no significant differences between them; however, in passive mobility in the sagittal plane, the results in the MT group were better. Therefore, changes in mobility from baseline to 12 weeks were minor and non-significant in either group. These results coincide with the findings of our study, where the neck mobility improved in a significant way only with manual techniques and stretching and isometric exercises and fulfillment of postural skills at home, especially in the flexion and extension movements; however, in the present study, significant differences in the MT group were maintained at 6-month follow-up.

Zaproudina et al. [13] compared three treatments for chronic neck pain: manipulative therapy, physical therapy (stretching, massage, and exercises), and only massage. They observed a significant improvement in neck mobility in the three space planes after 1 month of treatment in all three groups, finding no significant differences between the groups. It was found that neck spine mobility in rotation movements tended to improve more statistically after manipulative techniques than after massage. The present study also found significant differences in cervical mobility in the three space planes in the MT group post-intervention and at 6-month follow-up in the sagittal plane. In this way, thorough, non-manipulative manual techniques, with less risk and adverse effects for the patients, could improve mobility in the transversal plane.

In the study of Tuttle et al. [60], specific posterior-anterior mobilization techniques in the symptomatic cervical location were compared with three other groups (control, placebo, and non-manipulative manual therapy groups), finding a larger improvement in neck mobility with the non-manipulative manual therapy group than in the other groups.

In a clinical trial, Heikki and Hemmilä et al. [61] compared a group that received manipulative therapies combined with soft massage to a control group without therapy, finding a significant improvement in the treatment group only in the mobility in the three space planes (29% in FP, 23% in SP, and 16% in TP) after 5 weeks of treatment. In this study, the improvement post-intervention in the MT group was 6%, 8%, and 6.2%, respectively; however, we must consider that in Heikki´s study the patients´ mobility situation was worse at baseline, and patients paid 22 euros for five sessions, which had larger implications for the solution of their affliction.

In the same way, Korthals de Bos [39] concluded that spinal mobilization is more effective, and less expensive, for treating neck pain than physical therapy or medical treatment. On the other hand, Guzman et al. [3] stated that manipulation can produce a transitory increase in pain and discomfort in up to 30% of patients; this risk appears less likely with mobilization.

As regards the effects of TENS in the mobility of patients with neck pain, Nordermar et al. [62] compared three groups: neck collar, neck collar combined with MT, and neck collar combined with TENS in patients with acute cervical pain, not finding any statistically significant differences between the TENS and MT groups. This coincided with the results of our study.

On the other hand, Dusunceli et al. [63] found that TENS is effective for improving neck mobility in cervical pain if combined with other active therapies such as neck stabilization exercises. In the present study, the combination of TENS, active exercises, and postural hygiene did not improve cervical mobility, so it would be interesting to research the effects of stabilization exercises because TENS did not seem to show improvement in the patients. Some previous studies have also demonstrated endurance increases in the cervical musculature even after passive treatment such as massage and stretching [35,64,65,66]. Rehabilitation medicine offers a range of pain management approaches. The rehabilitative approach may be particularly helpful for patients with refractory movement-associated pain who do not want, or cannot tolerate, pharmaco-analgesia [67]. The need to integrate multiple rehabilitation and pain management treatments to enhance the results of each treatment is a very topical issue.

An increase in neck muscle endurance seems to be associated with a decrease in neck pain. In earlier studies, muscle pain has been reported to affect motor control, possibly leading to functional deficit [36,68,69]. Jordan et al. [65] suggested that the gain in strength in the subjects of their study was probably a result of increased confidence. Additionally, Al-Obaidi et al. [70] suggested that an improvement in the cognitive perception of pain, and fear-avoidance beliefs regarding physical activities, might contribute to the improvement of isometric muscle strength in patients with chronic pain.

According to the studies of Häkkinen et al. [46,71,72], it seemed that endurance training was the most effective treatment for strength recovery in patients with neck pain. In this way, in the systematic reviews, only limited evidence was found for the benefit of active strengthening exercises for neck disorders [9,73].

Chiu et al. [64] studied the effect of TENS associated with infrared irradiation, finding that after 6 weeks, patients showed significant improvement in their isometric neck muscle strength. These results did not coincide with our study; this important difference might be due to the fact that in Chiu’s study, TENS was applied over the neck acupuncture points. Recently, Schabrun et al. found that interactive neurostimulation, a technique similar to TENS, did not obtain significant improvements in functionality compared to a placebo in patients with MNP [72]. Further investigations are required to determine the effects of these techniques.

According to the indications of the PRECIS tool [74], the design of the study was realistic. The patients all had chronic/subacute neck pain, and all were recruited at primary healthcare physical therapy units where all were treated. Both interventions were usually used and delivered by physical therapists in the usual practice for the care of such patients. Moreover, mobility and endurance outcomes both had meaning to the patients.

On the other hand, the difficulties encountered during our investigation were described in a previous publication [21]. In the present study, the Palmer and Epler test was used as a measuring instrument to assess the endurance variable; however, most studies that assessed muscle strength [35,71,75] used an isometric neck strength testing machine described previously in reliable studies; other studies evaluated cervical muscle endurance with one of the most used tests: the skull cervical flexion test. The lack of uniformity in this variable and measuring instruments placed a limitation on comparisons of the results. Another important issue in this study was the variety of techniques considered as MT. This point made comparisons of the results with general studies more difficult [23,76].

As a matter of practical applicability, in the present study, only MT was proven to be useful in increasing active cervical mobility and endurance in patients with neck pain. Adverse effects from the application of this therapy are almost nonexistent and of little clinical importance. Therefore, MT is recommended for treating these patients. According to a recent systematic literature review and meta-analysis [77], manipulation and mobilization appear safe for treating chronic nonspecific neck pain. However, studies with much larger sample sizes would be required to fully describe the safety of manipulation and/or mobilization for this affliction.

As future research, it would be interesting to investigate whether improvements in isometric strength or endurance are more effective in terms of functionality and decreasing relapses in these patients [24,41]. Biomechanical overload depends on external demands and internal response capacity of the tissues, which affect functional disorders (lack of strength, deficit of mobility, posture disorder, etc.). If these functional changes are maintained, they can be memorized as new motor patterns and persist after the injury that generated them. Abnormal movements induce a major overload and can perpetuate it [42,43,44].

It would be necessary to conduct more trials as well as pain education sessions to gain a better understanding of patient satisfaction and expectations [78] and perform a cost-benefit and 6-month follow-up analysis [79,80] to provide physical therapists with a stronger basis for their decision-making process. In this regard, Guzman J. et al. [3] suggested that patients should be informed about the risks and benefits of the different treatments, and physical therapists should take into consideration patients´ preferences regarding their treatment techniques. It is essential that future research on the effectiveness of different techniques, and of MT in particular, specify the localization, dose, and description of the selected therapy.

## 5. Conclusions

Only MT had proven to be useful to increase active cervical mobility in the three planes and endurance in patients with neck pain. The adverse effects of the application of this therapy are almost nonexistent and have little clinical significance. Therefore, MT is recommended for treating patients with neck pain.

## Figures and Tables

**Figure 1 jcm-10-03245-f001:**
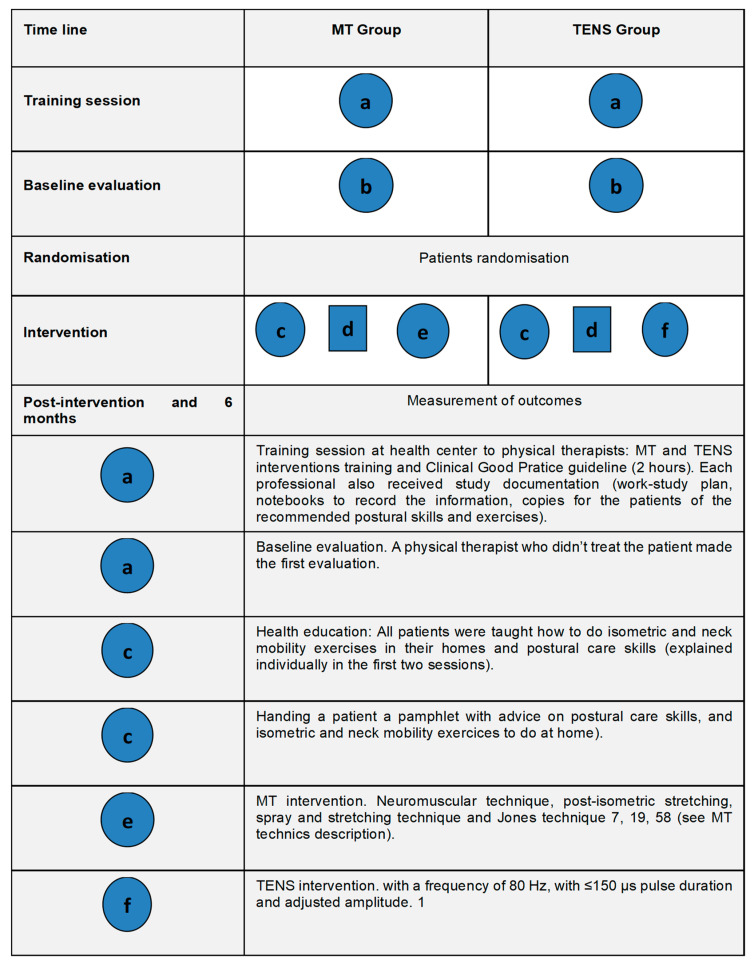
Graphical depiction of MT and TENS interventions. MT: Manual therapy; TENS: transcutaneous electrical nerve stimulation.

**Figure 2 jcm-10-03245-f002:**
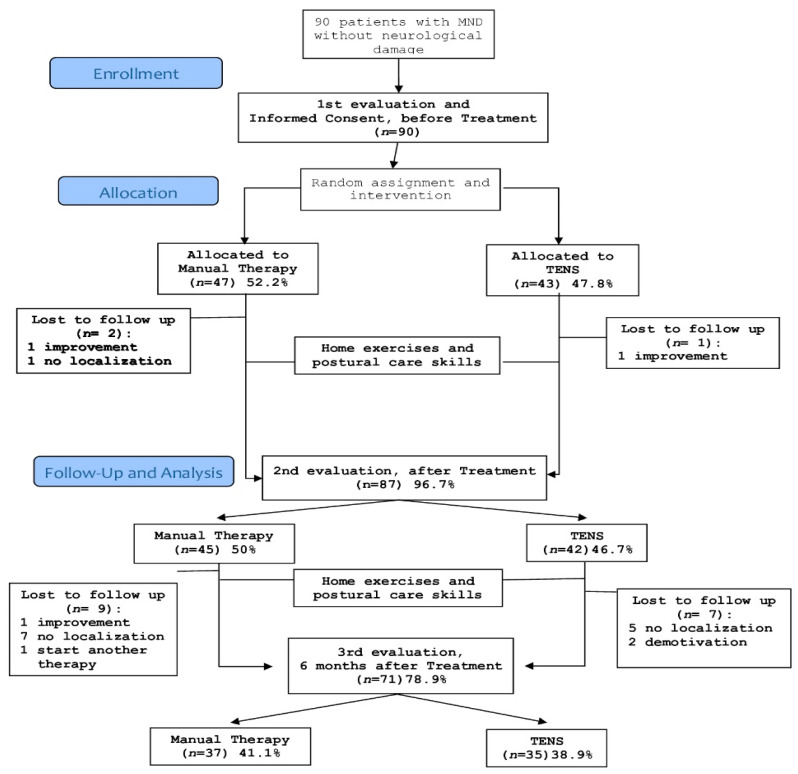
Flow chart: Progress of patients through trial.

**Figure 3 jcm-10-03245-f003:**
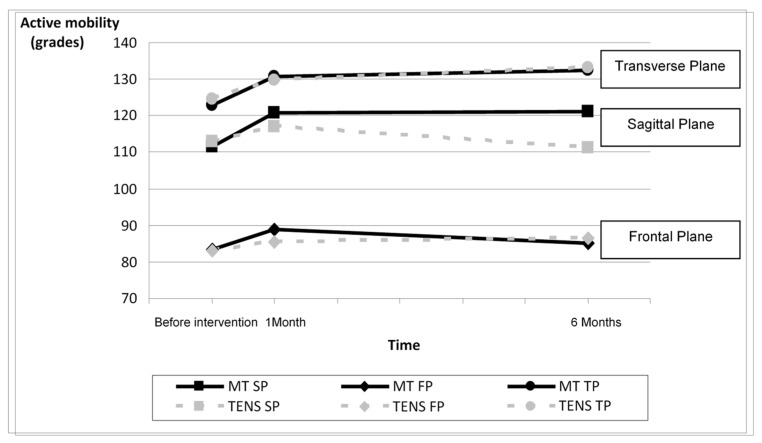
Active mobility evolution by planes in both intervention groups. Abbreviations: SP, sagittal plane; FP, frontal plane; TP, transverse plane; MT, manual Therapy; TENS, transcutaneous electric nerve stimulation.

**Figure 4 jcm-10-03245-f004:**
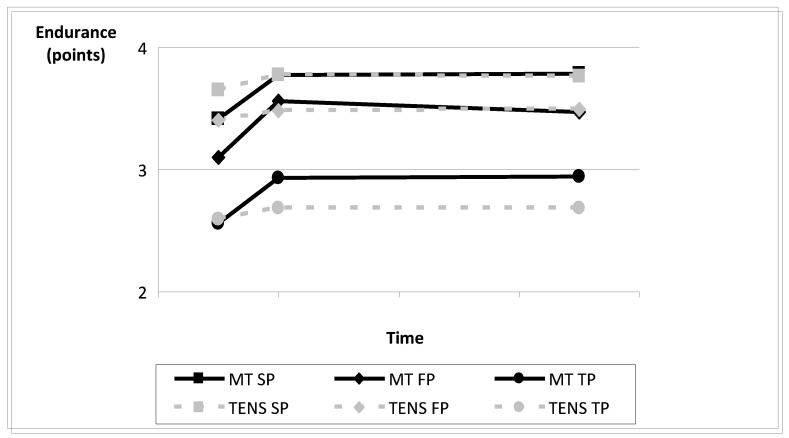
Endurance evolution by planes in both intervention groups. Abbreviations: SP, sagittal plane; FP, frontal plane; TP, transverse plane; MT, manual therapy; TENS, transcutaneous electric nerve stimulation.

**Table 1 jcm-10-03245-t001:** Description of Manual Therapy Technics [22,51,52].

MT Technic	Description
Neuromuscular technique	Longitudinal strokes were performed along the fibers of each muscle to stretch locally the areas of most tension that were previously located. The technique was applied with the physical therapist’s first finger, and using the last four fingers as support.
Post-isometric stretching	The patient was placed in a stretching position that was comfortable for the affected muscles, and he/she was asked to do an isometric contraction for 5 to 7 s. After that, a new barrier was searched for, and the sequence was repeated up to three times.
Spray and stretching technique	From a comfortable position for the patient, the first cold spray was applied to the target muscle’s MTrPs, in order to start immediately afterward with muscle stretching up to the position the patient allowed. After this, the area was sprayed again, but this time the whole area with pain was covered, in order to perform another stretch. On each area the spray could only be applied three times.
Jones technique	Pressure was applied to the painful points and/or the MtrPs, which had previously been located, until pain was generated. Then a position where pain disappeared was searched (maximum comfort position), and it was maintained for 90 s before passively returning to the original position.

Abbreviations: MT, manual therapy.

**Table 2 jcm-10-03245-t002:** Sociodemographic characteristics and the prognostic variables at baseline.

	Manual Therapy (*n* = 47)	TENS (*n* = 43)
Gender: women (*n* (%))	42 (89.4)	38 (88.4)
Age (decimal ages) (*X* (SD))	40.8 (11.6)	39.3 (9.7)
Previous neck pain episodes (*n* (%))	38 (84.4)	37 (86.0)
History of accident with cervical impaired (*n* (%))	12 (26.1)	6 (14.3)
Disability: NDI (*X* (SD))	31.6 (11.3)	34.4 (13.9)
Pain: VAS (*X* (SD))	54.9 (18.8)	56.4 (20.2)
Cervical current episode duration (days) (*X* (SD))	141 (280.8)	154.3 (216.2)

Abbreviations: *n* (%), number (percentage); *X* (SD), mean (standard deviation); NDI, neck disability index; VAS, visual analogue scale.

**Table 3 jcm-10-03245-t003:** Cervical active mobility and endurance differences before and after treatment in the different planes.

	Baseline		Basal and Post-Intervention			Basal and 6-Months Follow-Up		
	MT Group (CI 95%) (n = 47)	TENS Group *x* (CI 95%) (n = 43)	MT Group Difference (CI 95%) (n = 45)	TENS Group Difference (CI 95%) (n = 42)	Between Groups *p*-Value	MT Group Difference (CI 95%) (n = 36)	TENS Group Difference (CI 95%) (n = 35)	Between Groups *p*-Value
MOBILITY (planes)								
SAGITAL	111.62 (105.9–117.3)	112.86 (106.6–119.2)	8.93 (4.37–13.50)	3.44 (−1.54–8.42)	0.372 (*)	5.68 (0.37–11.0)	−2.38 (−11.37–6.60)	0.114 (*)
FRONTAL	83.34 (77.5–89.2)	82.95 (76.8–89.1)	4.98 (1.28–8.67)	1.62 (−3.70– 6.93)	0.583 (*)	−0.46 (−6.66–5.73)	3.09 (−4.45–10.62)	0.459 (*)
TRANSVERSAL	122.85 (115.2–130.5)	124.72 (116.2–133.3)	7.64 (1.24–14.05)	4.05 (-2.57– 10.67)	0.902 (*)	4.89 (−2.23–12.02)	8.79 (−0.39–17.97)	0.494 (*)
ENDURANCE (planes)								
SAGITAL	3.44 (3.2–3.6)	3.67 (3.5–3.8)	0.33 (0.15–0.49)	0.12 (−0.01–0.25)	0.140 (†)	0.3 (0.11–0.49)	0.11 (−0.04–0.28)	0.505 (†)
FRONTAL	3.11 (2.9–3.3)	3.42 (3.2–3.6)	0.43 (0.24–0.62)	0.08 (−0.13– 0.29)	0.114 (†)	0.3 (0.10–0.50)	0.07 (−0.19–0.34)	0.234 (†)
TRANSVERSAL	2.57 (2.3–2.7)	2.59 (2.3–2.7)	0.33 (0.12–0.55)	0.11 (−0.13–0.35)	0.117 (†)	0.32 (0.07–0.57)	0.04 (−0.24–0.33)	0.114 (†)

Abbreviations: *X*, mean; CI, confidence interval; MT, manual therapy; TENS, transcutaneous electrical nerve stimulation. *p <* 0.05. * General linear model of repeated means, simple contrast; † U de Mann–Whitney.

**Table 4 jcm-10-03245-t004:** Influence of study variables on cervical active mobility and endurance.

		Models		Intervention		Disability (NDI)		Episode Duration		Postural Compliance		Previous Episode	
		Rc	*p*	β	*p*	β	*p*	β	*p*	β	*p*	β	*p*
**Post-intervention**													
**MOBILITY (planes)**	SAGITAL	0.16	0.00	−6.58	0.03			−0.02	0.02	6.75	0.03	−9.63	0.04
	FRONTAL	0.04	0.07	−3.18	0.32					6.54	0.04		
	TRANSVERSAL	0.06	0.03	−5.26	0.24	0.46	0.01						
**ENDURANCE (planes)**	SAGITAL	0.09	0.02	0.84	0.01	−0.03	0.04						
	FRONTAL	0.15	0.00	0.29	0.07								
	TRANSVERSAL	0.06	0.02	0.25	0.11								
**Six months after intervention**													
**MOBILITY (planes)**	SAGITAL	0.10	0.01	−10.08	0.04	0.56	0.01						
	FRONTAL	0.08	0.04	3.58	0.44	0.42	0.04	−0.02	0.07				
	TRANSVERSAL	0.17	0.00	0.73	0.89	0.88	0.00						
**ENDURANCE (planes)**	SAGITAL	0.17	0.01	0.56	0.00					−0.81	0.04	0.31	0.10
	FRONTAL	0.26	0.00	0.06	0.73	0.02	0.00						
	TRANSVERSAL	0.16	0.00	0.20	0.26			0.00	0.00				

Abbreviations: Rc, coefficient of determination; p, p-value; β, standardized coefficient; NDI, neck disability index. *p* < 0.05.

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
