# Peer review of "Efficacy of Manual Therapy and Transcutaneous Electrical Nerve Stimulation in Cervical Mobility and Endurance in Subacute and Chronic Neck Pain: A Randomized Clinical Trial"

_jcm, 2021, doi:10.3390/jcm10153245_

Round 1

Reviewer 1 Report

This is an interesting, very well described and conducted randomized clinical trial about efficacy of manual therapy and TENS in cervical mobility and endurance in neck pain patients. However, some criticism should be raised:

Abstract

L20 - Try 'and' instead 'against'

L30 - 'produces' is not convenient English word in this context - look for replacement

Introduction

L76 - three dots does not look very scientific, try 'etc.' instead

In terms of the Introduction section, the Authors start with MT and TENS, while NSAID and DN are usually first-in-line, please incorporate an cite https://pubmed.ncbi.nlm.nih.gov/30984320/ as there are a lot of things in common here, hence neck and orofacial area are essentially intertwined in terms of referred pain.

Materials and methods

An entire RCT is very well defined and study design was published previously. Thus, it would be convenient to provide CONSORT diagram as well - this will improve clarity of patients enrollment/drop-out and eligibility.

L113 - 'Epidat' is missing manufacturers' data

L121-123 - 'NDI' abbreviation should be explained at L121 not L123

L149 - please provide valid manufacturers' data for TENS device - adhere to MDPI policy (device, manufacturer, city, state (if US or not clear), country)

L160 - provide details for CROM as there are none, there is a short mention present  in L167 but - as in my remaks above - it is not consistent with MDPI policy of referring to brand names and products

L217 - PSS© version 17.0  - as above

Discussion

L273-283 certain phrases should be moved to the bottom of entire section, possibly with separate subheading 'study limitations'

References

This section should be numbered and organized according to MDPI citation policy

Author Response

This is an interesting, very well described and conducted randomized clinical trial about efficacy of manual therapy and TENS in cervical mobility and endurance in neck pain patients.

RESPONSE#1 Thank you very much for your time and comments to help us improve our paper.

However, some criticism should be raised: Abstract L20 - Try 'and' instead 'against'

RESPONSE#2 Thank you very much for your kind suggestions, it has been changed.

L30 - 'produces' is not convenient English word in this context - look for replacement

RESPONSE#3 Thank you very much for your kind suggestions, it has been changed.

Introduction- L76 - three dots does not look very scientific, try 'etc.' instead

RESPONSE#4 Thank you very much for your kind suggestions, it has been changed.

In terms of the Introduction section, the Authors start with MT and TENS, while NSAID and DN are usually first-in-line, please incorporate an cite https://pubmed.ncbi.nlm.nih.gov/30984320/ as there are a lot of things in common here, hence neck and orofacial area are essentially intertwined in terms of referred pain.

RESPONSE#5 Thank you very much for your kind suggestions, it has been added.

Materials and methods: An entire RCT is very well defined and study design was published previously. Thus, it would be convenient to provide CONSORT diagram as well - this will improve clarity of patients enrollment/drop-out and eligibility.

RESPONSE#6 Thank you very much for your kind suggestions it has been added (Figure 2).

L113 - 'Epidat' is missing manufacturers' data

RESPONSE#7 Thank you very much for your kind suggestions it has been added.

L121-123 - 'NDI' abbreviation should be explained at L121 not L123

RESPONSE#6 Thank you very much for your kind suggestions it has been explained.

L149 - please provide valid manufacturers' data for TENS device - adhere to MDPI policy (device, manufacturer, city, state (if US or not clear), country) L160 - provide details for CROM as there are none, there is a short mention present  in L167 but - as in my remaks above - it is not consistent with MDPI policy of referring to brand names and products

RESPONSE#7 Thank you very much for your kind suggestions it has been added.

L217 - PSS© version 17.0 - as above

RESPONSE#8. Thank you very much for your kind suggestions it has been added.

Discussion: L273-283 certain phrases should be moved to the bottom of entire section, possibly with separate subheading 'study limitations'

RESPONSE#9. Thank you very much for your kind suggestions it has been changed.

References: This section should be numbered and organized according to MDPI citation policy

RESPONSE#10 Thank you very much for your kind suggestions the changes have been done.

Reviewer 2 Report

Best regards

Author Response

Dear authors,

Congratulations for Your work! The article is of scientific interest and in line with the aims of the journal. Moreover, the manuscript appears to be well established in its entirety and in each of its sections. So, I recommend just some minor revisions, as follow.

RESPONSE#1 Thank you very much for your time and comments to help us improve our paper.

- I suggest to replace the word “against” with the word “versus” in lines 20 and 87

RESPONSE#2 Thank you very much for your kind suggestions, it has been changed.

- I suggest to replace the word “chronification” with the word “chronicization” in line 74

RESPONSE#3 Thank you very much for your kind suggestions, it has been changed.

- the introduction is well laid out but can certainly be expanded upon, deepening in the first paragraph the theme of disability deriving from chronic pain of the spine as a whole, and in particular of the lumbar spine. In this sense You could find useful hints in the following article:

Farì G, Santagati D, Pignatelli G, Scacco V, Renna D, Cascarano G, Vendola F, Bianchi FP, Fiore P,

Ranieri M, Megna M. Collagen Peptides, in Association with Vitamin C, Sodium Hyaluronate,

Manganese and Copper, as Part of the Rehabilitation Project in the Treatment of Chronic Low Back

Pain. Endocr Metab Immune Disord Drug Targets. 2021 Feb 10. doi:

10.2174/1871530321666210210153619. Epub ahead of print. PMID: 33568038;

RESPONSE#4 Thank you very much for your kind suggestions, related to that we have added information about the papers and the references in the main text.

- Also the discussion is well structured and explains the effectiveness of cervical manipulations as a treatment for neck pain. In Your paper, two very interesting concepts are only partially mentioned and would deserve to be partly deepened to make the paper even more interesting. The one is the controversial evidence about the efficacy of TENS, with respect to which I suggest this review:

Paolucci T, Agostini F, Paoloni M, de Sire A, Verna S, Pesce M, Ribecco L, Mangone M, Bernetti A,

Saggini R. Efficacy of TENS in Cervical Pain Syndromes: An Umbrella Review of Systematic

Reviews. Applied Sciences. 2021; 11(8):3423. https://doi.org/10.3390/app11083423; the other one is the dissertation on the need to integrate multiple rehabilitation and pain management treatments to enhance the results of each treatment (line 351 et seq.). This is also a very topical issue for which I suggest the following source:

Cheville AL, Smith SR, Basford JR. Rehabilitation Medicine Approaches to Pain Management.

Hematol Oncol Clin North Am. 2018 Jun;32(3):469-482. doi: 10.1016/j.hoc.2018.02.001. Epub 2018 Mar17. PMID: 29729782.

RESPONSE#5 Thank you very much for your kind suggestions, related to that we have added information about the papers and the references in the main text.

- to conclude, a brief passage about the possible side effects of cervical manipulations is advisable. In this sense, maybe You could find useful the following meta-analysis:Coulter ID, Crawford C, Vernon H, Hurwitz EL, Khorsan R, Booth MS, Herman PM. Manipulation and Mobilization for Treating Chronic Nonspecific Neck Pain: A Systematic Review and MetaAnalysis for an Appropriateness Panel. Pain Physician. 2019 Mar;22(2):E55-E70. PMID: 30921975;PMCID: PMC6800035.

RESPONSE#6 Thank you very much for your kind suggestions, related to that we have added information about the papers and the references in the main text.

I renew the compliments for the great work and I wish you good luck.

RESPONSE#7 Thank you again for your kind suggestions.

Round 2

Reviewer 1 Report

All of my remarks were successfully addressed. I believe the manuscript greatly improved, hence it is suitable for publication.